# Determination of Three Typical Metabolites of Pyrethroid Pesticides in Tea Using a Modified QuEChERS Sample Preparation by Ultra-High Performance Liquid Chromatography Tandem Mass Spectrometry

**DOI:** 10.3390/foods10010189

**Published:** 2021-01-18

**Authors:** Hongping Chen, Xinlu Wang, Pingxiang Liu, Qi Jia, Haolei Han, Changling Jiang, Jing Qiu

**Affiliations:** 1Institute of Quality Standards and Testing Technology for Agro-Products, Chinese Academy of Agricultural Sciences, Key Laboratory of Agri-Food Quality and Safety, Ministry of Agriculture and Rural Affairs, Beijing 100081, China; thean27@tricaas.com (H.C.); 82101181095@caas.cn (X.W.); liupingxiangrti@outlook.com (P.L.); Jiaqi@caas.cn (Q.J.); 2Graduate School of Chinese Academy of Agricultural Sciences, Beijing 100081, China; Hanhaolei@tricaas.com (H.H.); jiangchangling@tricaas.com (C.J.); 3Tea Research Institute, Chinese Academy of Agricultural Sciences, Hangzhou 310008, China

**Keywords:** tea, pyrethroid pesticide metabolite, ultra-high performance liquid chromatography tandem mass spectrometry, modified QuEChERS

## Abstract

Pyrethroid pesticides are widely used on tea plants, and their residues of high frequency and concentration have received great attention. Until recently, the residues of typical metabolites of pyrethroid pesticides in tea were unknown. Herein, a modified “quick, easy, cheap, effective, rugged and safe” (QuEChERS) method for the determination of three typical metabolites of pyrethroid pesticides in tea, using ultra performance liquid chromatography tandem mass spectrometry, was developed. The mixture of florisil, octadecylsilane, and graphite carbon black was employed as modified QuEChERS adsorbents. A Kinetex C18 column achieved good separation and chromatographic peaks of all analytes. The calibration curves of 3-phenoxybenzoic acid (3-PBA) and 4-fluoro-3-phenoxybenzoic acid (4-F-3-PBA) were linear in the range of 0.1–50 ng mL^−1^ (determination coefficient R^2^ higher than 0.999), and that of *cis*-3-(2-chloro-3,3,3-trifluoroprop-1-en-1-yl)-2,2-dimethylcyclopropanecarboxylic acid (TFA) was in the range of 1–100 ng mL^−1^ (R^2^ higher than 0.998). The method was validated and recoveries ranged from 83.0% to 117.3%. Intra- and inter-day precisions were lower than or equal to 13.2%. The limits of quantification of 3-PBA, 4-F-3-PBA, and TFA were 5, 2, and 10 μg kg^−1^, respectively. A total of 22 tea samples were monitored using this method, and 3-PBA and TFA were found in two green tea samples.

## 1. Introduction

It is estimated that 2.4 million tons of active pesticide ingredients are used annually, mainly in agriculture [1]. A more significant estimate indicates that pesticides are generally persistent; about half of the detected substances have already been eliminated, and another 10% to 20% are stable transformation products [2]. The issue of pesticide transformation products is important because they leave the active moiety intact, generate toxicologically more potent structures, and have smaller molecular mass and more polarity than their parent compounds, which increases their solubility and perdurability. The transformation products and/or metabolites of pesticides in crops may be due to impurities in formula products applied in the field, or the results of abiotic transformation and plant metabolism. Pesticides with their transformation products and metabolites enter the food chain and then are exposed to human beings, resulting in chronic and acute poisoning.

Pyrethroid pesticides with low mammalian toxicity have been used worldwide since the 1980s as a replacement for the high toxic class of pesticides, such as organophosphorus and organochlorine compounds [3]. Pyrethroid pesticides have become one of the most important classes of insecticides in crops, and they have the largest market share, accounting for 38% in 2015 [4]. In China, pyrethroid pesticides were the only increased pesticide class in terms of use, with annual demands in 2016 approaching approximately 3800 tons [5]. Pyrethroids such as bifenthrin, λ-cyhalothrin, cypermethrin, deltamethrin, fenpropathrin, fenvalerate, permethrin, and cyfluthrin provide effective insecticide control for crop protection. The mass application of pyrethroid pesticides in agricultural crops inevitably involves the issue of their residues in crops and related foods. Li et al. found that 378 of 1450 fruit samples contained pyrethroid residues in the range of 0.005 to 1.2080 mg kg^−1^ [6]. High frequencies for pyrethroids, such as 63.4% for bifenthrin (ND-3.848 mg kg^−1^), 55.4% for λ-cyhalothrin (ND-3.244 mg kg^−1^), 46.5% for cypermethrin (ND-0.499 mg kg^−1^), and 24.8% for fenvalerate (ND-0.217 mg kg^−1^), were found in 101 tea samples [7]. Morgan et al. investigated 8 pyrethroid residues in repeated duplicated-diet solid food samples of 50 adults during the period of 2009–2011, and found that at least one pyrethroid or pyrethroid degradate was found in 49% and 2% of the monitored samples, respectively [8]. Due to the neurotoxicity, hepatotoxicity, development toxicity, and digestive system toxicity of pyrethroid pesticides for animals and human beings, such high residues at the mg kg^−1^ level in food present a healthy risk [9]. European Food Safety Authority (EFSA) formulated a maximum residue level of cypermethrin in the range of from 0.05 mg kg^−1^ to 2 mg kg^−1^, based on the risk assessment, and cypermethrin residues in some monitored food samples reported by previous studies exceeded the maximum residue levels (MRLs) [10].

Pyrethroids are readily transformed into intermediate products and metabolites by hydrolysis and photolysis. As shown in Figure 1, 3-phenoxybenzoic acid (3-PBA) can be formed from several pyrethroid pesticides, such as esfenvalerate, cypermethrin, deltathrin, permethrin, and cyhalothrin, which are widely used in tea gardens. *Cis*-3-(2-chloro-3,3,3-trifluoroprop-1-en-1-yl)-2,2-dimethylcyclopropanecarboxylic acid (TFA) can be transformed from bifenthrin and cyhalothrin, while 4-fluoro-3-phenoxybenzoic acid (4-F-3-PBA) can be formed from cyfluthrin [3]. Hydrolysis is an important degradation pathway for pyrethroid pesticides, highlighting that hydrolysis intermediates could be formed in the field or surfaces of agricultural crops. Following the disappearance of pyrethroid pesticides, 3-PBA was concurrently formed [11]. Although pyrethroid pesticides are used in tea gardens and the issue of their residues is concerning, their transformation products and metabolites have not been monitored and assessed, except in a paper reported by Mortimer et al. [12].

The potential adverse effects of 3-PBA on human beings and environments should be further considered, and knowledge regarding the dissipation pattern of 3-PBA in agricultural crops and environments needs to be addressed. Wang et al. found that 3-PBA inhibited the phagocytic ability of macrophages, increasing the reactive oxygen species level. These results illustrated that 3-PBA has an immunotoxic effect on macrophages [13]. 3-PBA showed antiestrogenic, antiandrogenic, and thyroid hormone receptor antagonistic activities [14]. Considering the toxicity of pyrethroids transformation products and metabolites, an analytical technique for these compounds in foods should be developed for the determination of their residues in foods and further risk assessment.

Several techniques have been developed for the determination of pyrethroid transformation products and metabolites using gas chromatography tandem mass (GC-MS/MS) [15], ultra-high performance liquid chromatography (UHPLC) [16], and UHPLC coupled with tandem mass spectrometry (UHPLC-MS/MS) [17] or high-resolution orbitrap mass spectrometry (UHPLC-Orbitrap MS) [18]. Due to the high polarity and low volatility of pyrethroids transformation products and metabolites, these compounds were derived with bis(trimethylsilyl) trifluoroacetamide (BSTFA) prior to GC-MS/MS analysis [19]. The UHPLC-MS/MS technique provided clear advantages for the analysis of transformation products and metabolites attributing to high selectivity, sensitivity, and accuracy. Although several approaches for the analysis of pyrethroid metabolites have been developed using UHPLC-MS/MS, these analytical protocols focused on environment water [17], soil [20], human urine, and plasma [21,22]. There are few methods for the determination of pyrethroid metabolites in agricultural crops and foods. For some simple matrices, such as human plasma and water [17], sample preparation did not involve clean-up [22]. However, for complex samples, such as soil [20], urine [21], wastewater [23], vegetables and fruits [24], and other substrates, it is necessary to carefully optimize the adsorbents and their dosage of dispersive solid-phase extraction (d-SPE) or SPE cartridges to obtain satisfactory purification effect. Therefore, the development of the UHPLC-MS/MS method for the complicated matrix is different from other current methods due to the optimization of sample preparation techniques.

Tea is a complex matrix, rich in tea polyphenols, caffeine, polysaccharides, and other polar substances, which greatly interferes with the analysis of polar compounds, like pyrethroid pesticide metabolites. In this study, we developed a sensitive method based on UHPLC-MS/MS for the determination of typical pyrethroid metabolites in tea samples. UHPLC-MS/MS parameters, such as chromatographic column and mobile phase, were optimized, and three modified “quick, easy, cheap, effective, rugged and safe” (QuEChERS) techniques, i.e., original QuEChERS [25], Official CEN 15662 [26], AOAC 2007.1 [27], were evaluated, and the adsorbents were optimized step by step.

## 2. Experimental

### 2.1. Reagents and Materials

3-PBA (purity 99%), 2-phenoxybenzoic acid (2-PBA) (purity 99%), 4-F-3-PBA (purity 99%), and TFA were obtained from Dr.Ehrenstorfer (Augsburg, Germany). HPLC-grade methanol and acetonitrile were purchased from Sigma-Aldrich (Shanghai, China). A Milli-Q-Plus system (Millipore, Milford, DE, USA) was used to obtain de-ionized water. Anhydrous magnesium, sodium chloride, dehydrate citrate sodium, sodium acetate, and disodium monohydrogen citrate were purchased from Shanghai Lingfeng Chemical Reagent Co., Ltd. (Shanghai, China). LC-MS-grade acetic acid and formic acid were obtained from Sigma-Aldrich (Shanghai, China). Amino carbon nanotubes (NH_3_-NTs), multi-walled carbon nanotubes (MWCNTs), graphite carbon black (GCB), primary secondary (PSA), octadecylsilane (C18), florisil, zirconia (ZrO), and polyvinylpyrrolidone (PVPP) were purchased from Bonna-Agela Technologies (Tianjin, China). Organic green tea and black tea, which were free of pesticide residues and pyrethroid pesticide metabolites, were supplied by the Key Laboratory of Tea Quality and Safety, Ministry of Agriculture. Commercial tea samples, including green tea (12) and black tea (10), were purchased from a supermarket, a whole market, and a monopoly shop located in Zhejiang province.

### 2.2. Sample Preparation

A 2.0 g portion of previously grounded tea was weighed into a 50 mL centrifuged tube. An internal standard, named 2-PBA, was added, and its concentration was 50 μg kg^−1^ by adding 100 μL of 2-PBA standard solution at 1 μg mL^−1^. The internal standard fortified in tea samples was used to calibrate the loss of target compound during sample preparation. For the matrix-matched standard solutions, the internal standard 2-PBA was added to the concentration at 10 μg L^−1^. Then, 2 mL of water was added and the tea sample was mixed on a vortex for 1 min. After that, 10 mL acetonitrile containing 1% acetic acid was used to extract the target compounds using an oscillator for 10 min. Then, 1.0 g of dehydrate citrate sodium and 0.5 g of disodium monohydrogen citrate were added to the mixture. The mixture was homogenized on a vortex for 1 min and then centrifuged at 10,000 rpm for 10 min. Following this, 2 mL of supernatant was transferred into a 5 mL centrifuge tube containing 200 mg florisil, 200 mg C18, and 100 mg GCB. The mixture was mixed on a vortex for 1 min and then centrifuged at 10,000 rpm for 10 min. The supernatant was filtered through a 0.22-μm membrane into the LC vial for UHPLC-MS/MS analysis.

### 2.3. UHPLC-MS/MS Analysis

Instrumental analysis was performed using an API 6500 triple quadrupole instrument mass spectrometry (AB SCIEX, Foster City, CA, USA) coupled to a Sciex 4000 UHPLC system (AB SCIEX, USA), both from Sciex Corporation (Foster City, CA, USA). Chromatographic separations employed a Kinetex C18 column (2.1 × 100 mm, particles 2.6 μm) (Phenomenex Co., Ltd, California, USA). A 10 μL sample loop was used for injection. The injection volume was 2 μL. The mobile phase consisted of water (A) and acetonitrile (B). Gradient elution was as follows: initial to 1 min, 5% B; 1–3 min, 95% B; 3–10 min, 95% B; 10–10.5 min, 5% B; 10.5–12 min, 5% B. UHPLC-MS/MS was performed in negative polarity (−4500 V). The source temperature was 550 °C. Gas 1, gas 2, curtain gas, and collision gas were set for nitrogen, and the flow rate was 60, 60, 30, and 10 psi, respectively. Scheduled multi reaction monitoring (sMRM) mode was employed for detecting the target compounds, and sMRM parameters are shown in Table 1.

### 2.4. Validation Procedure

The method was validated in accordance with the SANTE/11813/2017 standard, in terms of linearity, matrix effect, sensitivity, accuracy, and precision [28]. Linearity was evaluated using a matrix-matched standard solution at seven levels in the range of 0.1–200 μg L^−1^ in triplicates. Matrix matched standard solutions were prepared using the same type of analyzed tea samples, and internal standard 2-PBA was added to the concentration at 10 μg mL^−1^. Matrix effect (ME) was evaluated comparing the slope of the calibration curve obtained from the solvent and matrix-matched standard solution in the same levels, according to the following equation: ME (%) = (slope of matrix-matched calibration curve/slope of solvent calibration curve-1) × 100. The limits of detection (LOD) and quantification (LOQ) were estimated from the UHPLC-MS/MS signal through the lowest spiked level. LOD was defined as the lowest spiked concentration that provided a signal to noise ratio of 3, and LOQ was defined as the lowest spiked concentration with recovery in the range of 70 – 130% with relative standard deviations (RSDs) below 20%. Recoveries were calculated by comparing the measured values and spiked concentrations. Accuracy was expressed as recovery and assessed by the detecting concentrations in blank tea samples spiked with 10, 50, and 100 μg kg^−1^. Precision was estimated as intraday and interday variability by recovery experiment of three replicates in three days.

## 3. Results and Discussion

### 3.1. UHPLC-MS/MS Optimization

The MRM parameters were optimized by direct injection of individual standard solution at 0.5 μg mL^−1^ through a syringe pump at the flow rate of 10 μL min^−1^. Regarding analytes as acid compounds, ESI in negative mode was employed, and their deprotonated molecular ions [M − H]^−^ were acquired by full scan in the mass range of m/z 50–300. Each deprotonated molecular ion of three target compounds and internal standard could achieve two transitions. The ion transition with a higher signal was used for quantification, and another transition was used for confirmation. The isomers of 3-PBA and 2-PBA had the same transitions. The declustering potential and collision energy were optimized to improve sensitivity. Table 1 shows MRM parameters for each compound.

Several LC columns, such as Luna C8 (4.6 × 150 mm, 5μm) [22], XSELECT™ CSH™ C18 (100 × 2.1 mm id, 2.5 μm particle size) [23], the Betasil C18 column (100 × 2.1 mm id, 3 μm particle size) [29], Inspire C18 (25 cm × 4.6 mm, id 5 μm particle size) [17], and the Zorbax Eclipse Plus C18 column (100 × 2.1 mm id, 1.8 μm particle size) [18], have been used for the separation of pyrethroid metabolites. However, there is no report for investigating the effect of LC columns and mobile phase on the separation of target compounds. In this study, seven octadecylselyl-based columns, e.g., Zorbax Elipse Plus C18 (150 × 3.0 mm id, 1.8 μm particle size), Zorbax SB-Aq (100 × 2.1 mm id, 2.7 μm particle size), Poroshell 120 EC-C18 (100 × 2.1 mm id, 2.7 μm particle size), Aquity UHPLC HSS T3 (100 × 2.1 mm id, 1.8 μm particle size), Zorbax Eclipse XDB-C18 (150 × 3.0 mm id, 1.8 μm particle size), Kinetex C18 (50 × 2.1 mm id, 2.6 μm particle size) and Kinetex C18 (100 × 2.1 mm id, 2.6 μm particle size), were employed for the separation of target compounds. The results are shown in Figure 2 and Appendix A. Chromatographic peak broadening for the front four columns was observed, and poor separation of isomers 3-PBA and 2-PBA on Zorbax Eclipse XDB-C18 occurred. A good chromatographic peak was obtained on the Kinetec C18 column due to its core–shell particles with high peak capacity and resolution ability [30]. However, the isomers of 3-PBA and 2-PBA could not be separated on the shorter column with a length of 50 mm.

The mobile phase was also investigated. Compared with methanol, acetonitrile had high elution ability, and better chromatographic peaks for all compounds were observed. Acid water with 0.1% formic acid resulted in no retaining on Kinetex column, poor separation of 3-PBA and 2-PBA, and mass spectrometric signal decreasing eight times. Therefore, a Kinetex C18 column with mobile phase using water and acetonitrile was employed for the separation of target compounds.

### 3.2. Modified QuEChERS Development

Three extraction methods, namely, original QuEChERS [25], Official CEN 15662 [26], and AOAC 2007.1 [27], were evaluated. As shown in Figure 3, the highest recoveries of all analytes for CEN 15662 treatment were obtained among three extraction methods. Acidified acetonitrile was generally used for the optimal extraction of acid compounds to achieve high extraction efficiency [31,32]. In this study, the analytical compounds and internal standard 2-PBA were acid compounds with log P less than 4.0. Therefore, the citrate-buffering salts employed for the Official CEN 15662 method helped to improve the recoveries of target compounds when compared with the original QuEChERS and AOAC 2007.1 method, without using acidified acetonitrile.

The selection of QuEChERS adsorbents is crucial for the recovery of analytes and the removal of the tea matrix. Several adsorbents, such as PSA, C18, GCB, florisil, and MWCNTs, have been employed for cleaning up the tea matrix [33,34,35]. In this study, seven adsorbents, e.g., NH_3_-NTs, MTCNTs, GCB, C18, PSA, florisil, and ZrO, were investigated. Two milliliters of tea extracts spiked with three target compounds and internal standard at 100 μg L^−1^ was added in a 5 mL centrifuged tube individually containing NH_3_-NTs (100 mg), MTCNTs (100 mg), GCB (100 mg), C18 (200 mg), PSA (200 mg), florisil (200 mg), and ZrO (200 mg). The results are shown in Table 2. Satisfied recoveries of all target compounds were achieved in the range of 92.5–108.3% for five adsorbents, except PSA and PVPP, in which recoveries ranged from 55.7% to 111.9%, and from 86.4–97.9%, respectively. As shown in Figure 4, tea extracts became light yellow and transparent after they were cleaned up by three carbon materials, such as NH_3_-NTs, MWCNTs, and GCB, because these adsorbents were particularly effective for tea co-extracted pigments [36]. Tea extracts were dark and opaque before and after clean up by ZrO, indicating that a few tea matrices were removed. 

Although the decolorization effect of C18 and florisil was not obvious, considering the fact that C18 and florisil were useful and employed to remove some mid-polar and high lipid contents [37,38], the mixture of 200 mg C18 and 200 mg florisil combination with individual MWCNTs and GCB was further evaluated as modified QuEChERS. As shown in Table 3, the mixture of MWCNTs, C18, and florisil resulted in the loss of 3-PBA, 2-PBA, and 4-F-3-PBA, while the combination of GCB with C18 and florisil achieved good recoveries of all target compounds in the range of 84.2 – 98.8%. Therefore, the mixtures of 100 mg GCB, 200 mg C18, and 200 mg florisil were used as modified QuEChERS adsorbents.

### 3.3. Method Performance

The linearity of the present method was evaluated using acetonitrile and matrix-matched calibration curves at seven different concentration levels in the range of 0.1 to 100 μg L^−1^ in triplicates at each level. 2-PBA at 10 μg L^−1^ was used as an internal standard for calibration of 3-PBA, 4-F-3-PBA, and TFA. The calibration curves of 3-PBA, 4-F-3-PBA, and TFA were obtained by plotting the peak area ratios against the concentrations. Satisfactory linearity of three analytes was obtained with determination coefficients (R^2^) higher than 0.998 in both acetonitrile and tea matrix-matched calibration solutions (Table 4).

Matrix effects (MEs) were evaluated by comparison of the slope ratio between matrix-matched calibration curves and acetonitrile calibration curves. All target compounds showed a strong matrix suppression effect, ranging from −53.2% to −83.7%. The black tea matrix resulted in a more serious suppression effect than that of the green tea matrix. Strong Mes were also observed for the determination of pyrethroids degradation products in human urine samples [18]. Therefore, the matrix-matched calibrations were used for quantification purposes, employing blank green tea samples and black tea samples.

Blank tea samples spiked at 1, 2, 5, and 10 μg kg^−1^ were determined for the evaluation of LODs and LOQs. As shown in Table 4, LODs of 3-PBA, 4-F-3-PBA and TFA were 1.9–2.2, 0.5 and 5.2–6.1 μg kg^−1^, respectively, while LOQs were 5, 2, and 10 μg kg^−1^ for 3-PBA, 4-F-3-PBA and TFA, respectively, in green tea and black tea samples. Mortimer et al. developed a sensitive gas chromatography electronic capture detector (GC-ECD)-based method for the determination of pyrethroid pesticides metabolites, where the LODs of 3-PBA and TFA were 5.0 and 1.4 μg L^−1^ [12]. Although the sensitivity of the previous study reported by Mortimer et al. was similar to this developed method, its complicated and time-consuming derivatization step was a shortage. Meanwhile, the LODs depended on the coextractive interferences because of the poor selectivity of GC-ECD.

Table 5 shows the intraday and interday accuracy and precision estimated through recovery trials, spiking blank samples at 10, 50, and 100 μg kg^−1^. Satisfactory recoveries were obtained in the range of 83.0–108.6% for 3-PBA, 82.9–92.3% for 4-F-3-PBA, and 107.8–117.3% for TFA in green tea and black tea. The precisions expressed as RSDs of 3-PBA, 4-F-3-PBA and TFA ranged from 2.4% to 13.2%, 1.8% to 9.2%, and 1.2% to 9.5% for intraday precision, while the interday RSDs of 3-PBA, 4-F-3-PBA and TFA ranged from 3.1% to 11.0%, 4.0% to 11.1%, and 2.2% to 8.0%, respectively. Typical UHPLC-MS/MS chromatograms of three target compounds and the internal standard are shown in Figure 5.

### 3.4. Analysis of Real Sample

The proposed method was applied to analyze 12 green tea samples and 10 black tea samples. Only two green tea samples were found to be contaminated with 3-PBA at 28.7 μg kg^−1^ and TFA at 12.9 μg kg^−1^ for one sample, and 3-PBA at 9.5 μg kg^−1^ for another sample. All black tea samples did not contain 3-PBA, 4-F-3-PBA, and TFA. To illustrate the formation of 3-PAB and TFA, we detected pyrethroid pesticides in both positive green tea samples referring to the previous method [39]. The results show that the positive sample with 3-PBA and TFA contained cypermethrin (649 μg kg^−1^), bifenthrin (1021 μg kg^−1^), and cyhalothrin (406 μg kg^−1^), while another positive green tea sample with 3-PBA only contained cypermethrin (31 μg kg^−1^). The results indicate that 3-PBA could be transformed from cypermethrin and cyhalothrin, while TFA was converted from bifenthrin and cyhalothrin.

Although pyrethroid pesticides were not found in the present black tea samples, 3-PBA and other metabolites could be produced during tea fermentation when pyrethroid pesticides were applied in the tea plant. Hu et al. found a novel golden flower fungus from fu brick tea, named *Eurotium cristatum* ET1 strain, which could efficiently degrade both cypermethrin and 3-PBA [40]. They also found that 3-PBA could be formed when ET1 degraded cypermethrin in brick tea, which was one of the heavy fermentation teas. Therefore, metabolites of pyrethroid pesticides could appear in both unfermented and fermented tea samples, and, thus, both pesticides and their metabolites should be monitored.

## 4. Conclusions

A modified QuEChERS method was developed for the determination of three typical pyrethroid metabolites, namely 3-PBA, 4-F-3-PBA, and TFA in tea using UHPLC-MS/MS. The UHPLC-MS/MS parameters were optimized and three versions of QuEChERS technique were compared. Various types of adsorbents were evaluated for obtaining high recoveries of target compounds and removal of tea matrices. Method validation was employed, and high recoveries and precisions were obtained, indicating the developed method fulfilled with the analysis of 3-PBA, 4-F-3-PBA, and TFA in tea.

## Figures and Tables

**Figure 1 foods-10-00189-f001:**
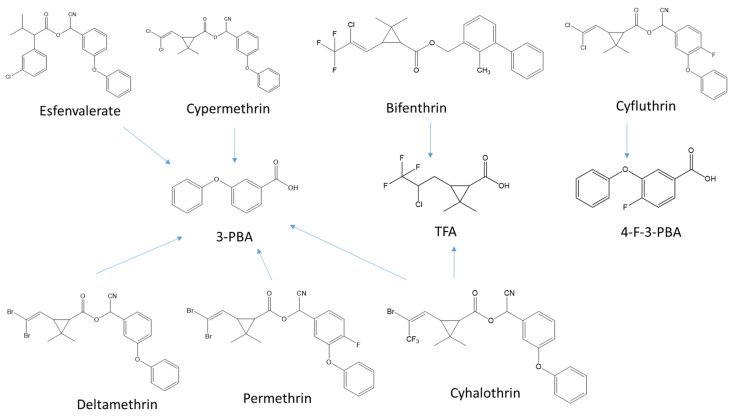
Transformation compounds from pyrethroids pesticides.

**Figure 2 foods-10-00189-f002:**
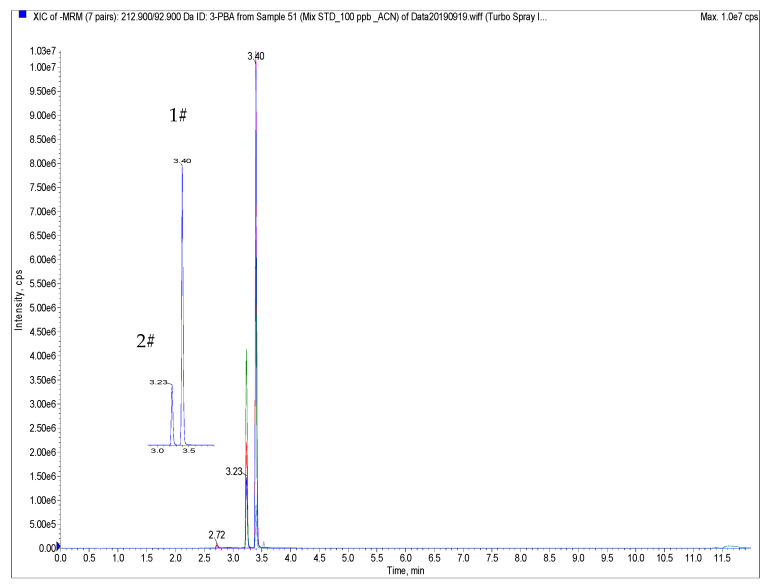
Ultra-high performance liquid chromatography coupled with tandem mass spectrometry (UHPLC-MS/MS) chromatograms of three target compounds and internal standard 2-PBA (peak 1 # means 3-PBA and peak 2# means 2-PBA) obtained from Kinetex C18 (100 × 2.1 mm id, 2.6 μm particle size) under the mobile phase A water and phase B methanol. Note, gradient elution: 0–1 min 5% B; 1–3 min 95% B; 3–10 min 95% B, 10–10.5 min 5% B;10.5–12 min 5% B.

**Figure 3 foods-10-00189-f003:**
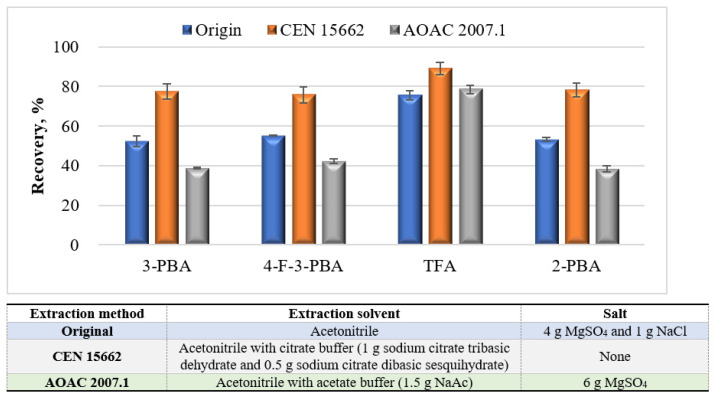
Recoveries of three analytes and internal substances with the extraction methods of original “quick, easy, cheap, effective, rugged and safe” (QuEChERS), CEN 15662, and AOAC 2007.1.

**Figure 4 foods-10-00189-f004:**
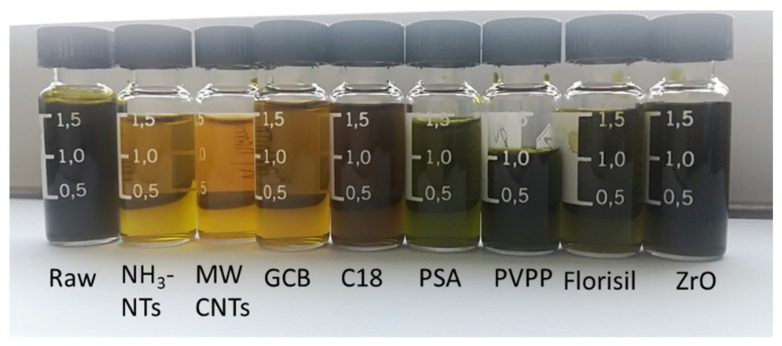
Tea extracts before (Raw) and after clean up by NH3-NTs, MWCNTS, GCB, C18, PSA, PVPP, florisil, and ZrO.

**Figure 5 foods-10-00189-f005:**
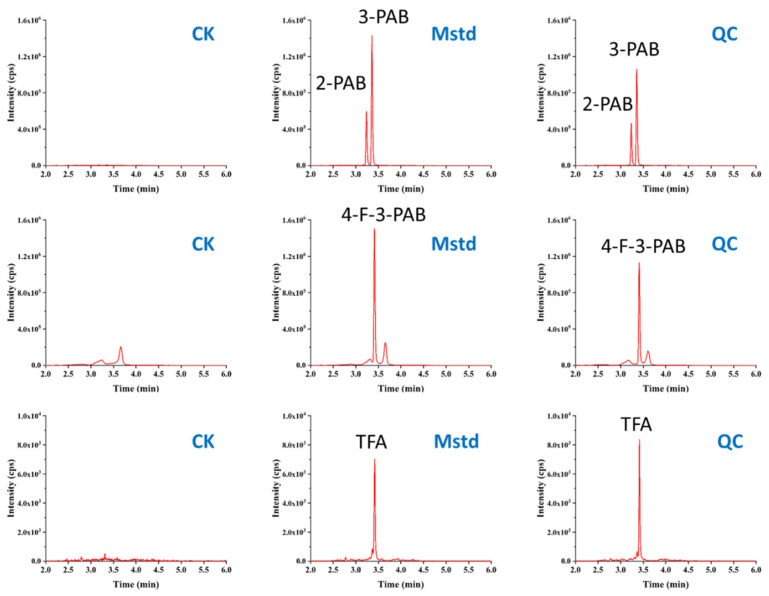
UHPLC-MS/MS chromatograms of 3-PBA, 4-F-3-PBA, TFA and internal standard 2-PBA in a blank green tea sample (CK), matched matrix standard calibration at 10 μg L^−1^ (Mstd), and spiked sample at 50 μg L^−1^ (QC).

**Table 1 foods-10-00189-t001:** MS/MS parameters for the analysis of 3-phenoxybenzoic acid (3-PBA), 4-fluoro-3-phenoxybenzoic acid (4-F-3-PBA), *cis*-3-(2-chloro-3,3,3-trifluoroprop-1-en-1-yl)-2,2-dimethylcyclopropanecarboxylic acid (TFA), and internal standard 2-phenoxybenzoic acid (2-PBA).

Compound	Retention Time (min)	Precursor (m/z)	Declustering Potential DP (V)	Production (m/z)	Collision Energy CE (eV)
3-PBA	3.35	212.9	−27	92.9	−27
				169.1	−18
4-F-3-PBA	3.31	230.9	−25	92.9	−33
				167.0	−25
TFA	3.450	240.9	−30	35.0	−15
				121.0	−15
2-PBA	3.19	212.9	−27	92.9	−27
				169.1	−18

**Table 2 foods-10-00189-t002:** Recoveries of target compounds after clean-up by amino carbon nanotubes (NH_3_-NTs), multi-walled carbon nanotubes (MWCNTS), graphite carbon black (GCB), octadecylsilane (C18), primary secondary (PSA), polyvinylpyrrolidone (PVPP), florisil, and zirconia (ZrO).

Adsorbents	Recoveries (%)			
	3-PBA	2-PBA	4-F-3-PBA	TFA
NH_3_-NTs	103.0 ± 2.4	101.6 ± 1.3	102.3 ± 4.2	103.8 ± 1.6
MWCNTs	103.2 ± 2.3	108.3 ± 1.8	103.1 ± 5.1	102.0 ± 7.8
GCB	99.8 ± 2.9	104.9 ± 1.4	99.7 ± 5.0	92.6 ± 1.9
C18	93.9 ± 3.0	103.6 ± 2.0	93.7 ± 6.8	93.8 ± 5.9
PSA	81.0 ± 3.7	55.7 ± 3.5	88.9 ± 2.8	111.9 ± 5.3
PVPP	86.4 ± 4.2	97.9 ± 0.9	88.4 ± 2.3	95.2 ± 2.5
Florisil	105.7 ± 3.9	98.4 ± 2.6	104.9 ± 2.7	104.9 ± 3.3
ZrO	94.0 ± 3.3	103.6 ± 0.7	94.1 ± 3.3	92.5 ± 4.4

**Table 3 foods-10-00189-t003:** The effect of different amounts of GCB and MWCNTs combined with 200 mg C18 and 200 mg florisil on the recovery of analytes.

Adsorbents	Recoveries (%)			
	3-PBA	2-PBA	4-F-3-PBA	TFA
GCB (25 mg) + C18 + Florisil	86.0	87.1	87.2	91.6
GCB (50 mg) + C18 + Florisil	84.2	86.0	86.8	97.6
GCB (75 mg) + C18 + Florisil	84.5	87.7	87.2	97.1
GCB (100 mg) + C18 + Florisil	85.6	87.5	89.0	98.8
MWCNTs (25 mg) + C18 + Florisil	82.0	82.4	82.9	92.0
MWCNTs (50 mg) + C18 + Florisil	78.1	76.4	79.7	91.9
MWCNTs (75 mg) + C18 + Florisil	75.7	78.8	76.7	99.0
MWCNTs (100 mg) + C18 + Florisil	69.4	75.8	74.5	92.0

**Table 4 foods-10-00189-t004:** Linear range and equation with correlation coefficient (R^2^), matrix effect (ME, %), the limit of detection (LOD, μg kg^−1^), and the limit of quantification (LOQ, μg kg^−1^).

Compounds	Matrix	Linear Rang (ng mL^−1^)	Equation	R^2^	MEs (%)	LOD (μg kg^−1^)	LOQ (μg kg^−1^)
3-PBA	Green tea	0.1–50	Y = 2.341x + 0.533	0.9994	−82.9	1.5	5
	Black tea	0.1–50	Y = 1.834x + 0.177	0.9999	−83.7	2.2	5
4-F-3-PBA	Green tea	0.1–50	Y = 5.426x + 0.628	0.9999	−60.3	0.5	2
	Black tea	0.1–50	Y = 3.810x + 0.330	1.0000	−63.6	0.5	2
TFA	Green tea	1–100	Y = 0.0263x + 0.0109	0.9993	−53.2	5.2	10
	Black tea	1–100	Y = 0.0147x + 0.0105	0.9984	−65.9	6.1	10

**Table 5 foods-10-00189-t005:** Recovery, relative standard deviations (RSDs), LOD, and LOQ were obtained for the target compounds in green tea (GT) and black tea (BT).

Compounds	Teas	Recoveries, %			Intra-Day Precision (RSD, %, n = 5)			Inter-Day Precision (RSD, %, n = 3)		
		Spiked Level (μg kg^−1^)			Spiked Level (μg kg^−1^)			Spiked Level (μg kg^−1^)		
		10	50	100	10	50	100	10	50	100
3-PBA	GT	108.6	83.0	95.4	4.6	3.1	2.4	11.0	6.0	10.3
	BT	101.0	100.1	99.7	13.2	3.1	5.4	7.9	3.5	3.1
4-F-3-PBA	GT	92.1	82.9	92.3	5.3	3.7	1.8	6.4	5.3	4.0
	BT	88.0	88.3	85.1	9.2	3.9	2.6	6.1	4.3	11.1
TFA	GT	110.4	113.4	109.7	3.6	1.2	4.9	8.0	4.0	6.1
	BT	117.3	112.0	107.8	9.5	1.6	3.2	5.1	2.6	2.2

## Data Availability

Not applicable.

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
