# Peer review of "Determination of Three Typical Metabolites of Pyrethroid Pesticides in Tea Using a Modified QuEChERS Sample Preparation by Ultra-High Performance Liquid Chromatography Tandem Mass Spectrometry"

_foods, 2021, doi:10.3390/foods10010189_

Round 1

Reviewer 1 Report

The manuscript by Chen and co-workers reports an optimization of analytical methods that can be used for the determination of three metabolites of widely used pyrethroids. Although it contains some meaningful results, it is a sort of technical report written by a contract analytical laboratory, not a scientific paper with sufficient novelty. The scientific merits do not satisfy the requirements of Foods in the reviewer’s eyes. A few specific comments that might be helpful are as follows:

Abstract: The novelty of manuscript should be emphasized. It is a summary of a technical report. If authors want to claim that their modification of QuEChERS is novel and significant, how the improvement was achieved should be stressed.

Line 33: “molecular mass”

Line 34: It is not always true because polar metabolites are often not as volatile as their parents.

Line 55: “can be formed”, not “converted”

Line 56: Italicize “cis”

Line 83-86: It is weird that UHPLC-MS-MS has been rarely applied for determination of pyrethroid metabolites. Authors already mentioned that they have used HPLC-MS-MS and UHPLC-MS-MS. It does not stress the novelty of this work.

Figure 1: Please adjust the aspect ratio.

Line 92: Please write the full name of “QuEChERS”.

Line 113: Please add details about the internal standard.

Figure 3: All three methods compared here have been published by others. The contribution of authors seems to be minimal.

Author Response

Review 1#

Comment: The manuscript by Chen and co-workers reports an optimization of analytical methods that can be used for the determination of three metabolites of widely used pyrethroids. Although it contains some meaningful results, it is a sort of technical report written by a contract analytical laboratory, not a scientific paper with sufficient novelty. The scientific merits do not satisfy the requirements of Foods in the reviewer’s eyes.

Response: Thanks for your comments. The novelty of this study was addressed in the revised manuscript.

Comment: Abstract: The novelty of manuscript should be emphasized. It is a summary of a technical report. If authors want to claim that their modification of QuEChERS is novel and significant, how the improvement was achieved should be stressed.

Response: The novelty of this study is to firstly develop a method for the determination of three typical metabolites of pyrethroid pesticides in tea, which of parent pesticides have been widely used in tea plants, resulting in high frequency and residues in tea. The novelty of this study was stressed in Abstract in the revised manuscript.

Comment: Line 33: “molecular mass”

Response: It has been revised. Thanks for your remind.

Comment: Line 34: It is not always true because polar metabolites are often not as volatile as their parents.

Response: The volatility has been replaced with perdurability, and corresponding reference was added in the revised manuscript.

Comment: Line 55: “can be formed”, not “converted”

Response: It has been revised. Thanks for your remind.

Comment: Line 56: Italicize “cis”

Response: It has been revised. Thanks for your remind.

Comment:  Line 83-86: It is weird that UHPLC-MS-MS has been rarely applied for determination of pyrethroid metabolites. Authors already mentioned that they have used HPLC-MS-MS and UHPLC-MS-MS. It does not stress the novelty of this work.

Response: As mentioned in Line 81-83, several methods based-UHPLC-MS/MS have been developed for determination of pyrethroid metabolites, but the reported methods were used for the analysis of pyrethroid metabolites in environment water, soil, human urine and plasma, which are different from the matrices of agricultural crops and foods. For UHPLC-MS/MS, different matrices require different sample preparing techniques due to matrix effect and instrument contamination by complicated matrices. However, there are few methods for the determination pyrethroid metabolites in agricultural crops and foods. The novelty is the complicated tea matrices and trace analysis of three typical pyrethroid pesticide metabolites in tea, which has been addressed in Instruction.

Comment: Figure 1: Please adjust the aspect ratio.

Response: It has been adjusted. Thanks for your remind.

Comment: Line 92: Please write the full name of “QuEChERS”.

Response: It has been revised. Thanks for your remind.

Comment: Line 113: Please add details about the internal standard.

Response: It has been revised. Thanks for your remind.

Comment:  Figure 3: All three methods compared here have been published by others. The contribution of authors seems to be minimal.

 Response: The three methods are official analytical methods for the determination of pesticide residues in food, but they have not been used and developed for the determination pyrethroid pesticide metabolites in tea.

Reviewer 2 Report

The topic of this paper is interesting and within the scopes of the journal. It is also generally well organized and written. Minor changes are suggested to improve the Materials & Methods section as well as the presentation of the results.

Line 83-86. Add the cites of these few papers and rephase to stress the novelty of the work.

Line109. Change “real” into “commercial”

Line 136. Add details about recovery calculation and preparation of the matrix-matched solutions.

Line 156. “confirmation”

Figure 2. Unclear. It is hard to see the peaks of the four compounds. Show only the chromatographic profile corresponding to the final conditions, the rest could be included as supplementary material.

Line 199. Spiked with the three compounds?   

Figure 6. Legend description too large. Move part of it into the text.

Line 275. “fungus from”

Author Response

The topic of this paper is interesting and within the scopes of the journal. It is also generally well organized and written. Minor changes are suggested to improve the Materials & Methods section as well as the presentation of the results.

Comment: Line 83-86. Add the cites of these few papers and rephase to stress the novelty of the work.

Response: Thanks for your suggestion. The references were added and introduced. The novelty of this study was presented in the last paragraph of Introduce.

Comment: Line109. Change “real” into “commercial”

Response: It has been revised. Thanks for your suggestion.

Comment:  Line 136. Add details about recovery calculation and preparation of the matrix-matched solutions.

Response: Recovery calculation and preparation of the matrix matched solutions have been added in the revised manuscript.

Comment: Line 156. “confirmation”

Response: It has been revised. Thanks for your kind help.

Comment: Figure 2. Unclear. It is hard to see the peaks of the four compounds. Show only the chromatographic profile corresponding to the final conditions, the rest could be included as supplementary material.

Response: It has been revised. Thanks for your suggestion.

Comment: Line 199. Spiked with the three compounds?   

Response: It was spiked with three compounds and internal standard. It has been revised.

Comment: Figure 6. Legend description too large. Move part of it into the text.

Response: It has been revised. Thanks for your suggestion.

Line 275. “fungus from”

Response: It has been revised. Thanks for your kind help.

Reviewer 3 Report

Title:  Determination of three typical metabolites of pyrethroid pesticides in tea using a modified QuEChERS sample preparation by ultra performance liquid chromatography tandem mass spectrometry

This paper reports the quantitative determination of three typical metabolites of pyrethroid pesticides in tea by UPLC-MS/MS. Many chromatographic methods for the analysis of pyretroids in food and biological matrices are reported in literature.

The work is well designed and validated. The method does not present great chromatographic innovation. The novelty concerns the choice of the investigated matrix (tea sample), the optimization of extraction method by modifying existing procedures and the application to 22 tea commercial samples.

Introduction

Lines 47-53: The authors report pyrethroids residues range in several food samples. It would be interesting to report the concentrations responsible for toxic/lethal effects so to make a comparison to data found in real cases.

Lines 62-64: Mortimer et al. report the quantification of pyrethroid metabolites in tea by gas chromatographic analysis (Analysis of total free and glucose-conjugated pyrethroid acid metabolites in tea infusions as hexafluoroisopropyl esters by gas chromatography with electron capture detection. Journal of AOAC INTERNATIONAL 1995, 78(3), 846–855). The authors should add this reference in the text and compare the sensitivity limits of the two methods.

Experimental

2.1 Reagents and Materials

Line 95: Replace “Regents” with “Reagents”

Line 113: Why was 2-PBA chosen as internal standard? Usually, an isotopically labelled internal standard is used in LC-MS/MS methods.

  1. Results and Discussion

3.1. UHPLC-MS/MS optimization

Line 156-172: As the authors themselves state, the analyte 3-PBA and the internal standard 2-PBA have the same transitions and show difficulties in chromatographic separation with the tested columns. The same column Kinetex C8, chosen for chromatographic separation, could not be able to separate the two isomers if it is used with length of 50 mm instead of100 mm. These drawbacks would have been eliminated with the use of an isotopically labelled internal standard.

Author Response

Review 3#

Title:  Determination of three typical metabolites of pyrethroid pesticides in tea using a modified QuEChERS sample preparation by ultra performance liquid chromatography tandem mass spectrometry

This paper reports the quantitative determination of three typical metabolites of pyrethroid pesticides in tea by UPLC-MS/MS. Many chromatographic methods for the analysis of pyretroids in food and biological matrices are reported in literature.

The work is well designed and validated. The method does not present great chromatographic innovation. The novelty concerns the choice of the investigated matrix (tea sample), the optimization of extraction method by modifying existing procedures and the application to 22 tea commercial samples.

Introduction

Comment:  Lines 47-53: The authors report pyrethroids residues range in several food samples. It would be interesting to report the concentrations responsible for toxic/lethal effects so to make a comparison to data found in real cases.

Response: Thanks for your suggestion. In the revised, the toxicity of pyrethroid pesticides were simply introduced and MRLs of cypermethrin formulated by EU were referenced.

Comment:  Lines 62-64: Mortimer et al. report the quantification of pyrethroid metabolites in tea by gas chromatographic analysis (Analysis of total free and glucose-conjugated pyrethroid acid metabolites in tea infusions as hexafluoroisopropyl esters by gas chromatography with electron capture detection. Journal of AOAC INTERNATIONAL 1995, 78(3), 846–855). The authors should add this reference in the text and compare the sensitivity limits of the two methods.

Response: This reference has been added. Thanks for your suggestion. The method reported by Mortimer et al. is also very sensitive, but the method was based on GC-ECD and the metabolites needed to be derived. Thus, this developed method is quicker, easier and simpler compared with the previous method. The revised manuscript compared the sensitivity and performance between this developed method and the previous method.

Experimental

2.1 Reagents and Materials

Comment: Line 95: Replace “Regents” with “Reagents”

Response: It has been revised. Thanks for your remind.

Comment: Line 113: Why was 2-PBA chosen as internal standard? Usually, an isotopically labelled internal standard is used in LC-MS/MS methods.

Response: Considering the similar physicochemical properties of target compounds with 2-PBA, 2-PBA was used as internal standard. We tried to purchase some isotopically labelled internal standard, but we failed. However, 2-PBA was a suitable internal standard for calibrating the loss during sample preparation, and good accuracy and precision were achieved for three target compounds.

  1. Results and Discussion

Comment: 3.1. UHPLC-MS/MS optimization

Line 156-172: As the authors themselves state, the analyte 3-PBA and the internal standard 2-PBA have the same transitions and show difficulties in chromatographic separation with the tested columns. The same column Kinetex C8, chosen for chromatographic separation, could not be able to separate the two isomers if it is used with length of 50 mm instead of100 mm. These drawbacks would have been eliminated with the use of an isotopically labelled internal standard.

Response: Thanks for your suggestion. Although the isotopically labelled internal standard was the best choice, the separation between 2-PBA and 3-PBA was good after the optimization of UPLC conditions. Thus, 2-PBA was used as internal standard. On the other hand, isotopically labelled standard was hard to obtain, and it could be much more expensive than 2-PBA. Sure, isotopically labelled standard is a better internal standard, and it dose not need to take much time for UPLC optimization.

Round 2

Reviewer 1 Report

Although the scientific merits are rather limited, authors improved the manuscript emphasizing the novelty of the work. It needs minor revisions.

Figure 2 caption: Do not use "um" for micro meters.

Recoveries in Figs 3, 4, 6 are better presented in Tables showing mean and standard deviation values clearly.

English needs to be polished:

e.g., "a sensitive based-GC-ECD method" -> "a sensitive GC-ECD-based method"

Author Response

Although the scientific merits are rather limited, authors improved the manuscript emphasizing the novelty of the work. It needs minor revisions.

Comment:  Figure 2 caption: Do not use "um" for micro meters.

Response: It has been revised. Thanks for your remind.

Comment:  Recoveries in Figs 3, 4, 6 are better presented in Tables showing mean and standard deviation values clearly.

Response: Recoveries in Figs 4, 6 are presented in Tables in the revised manuscript. Considering it is too simple when Fig 3 is presented in Table, Fig 3 did not change in the revised manuscript.

Comment: English needs to be polished:

e.g., "a sensitive based-GC-ECD method" -> "a sensitive GC-ECD-based method"

Response: We checked the manuscript carefully, and the typo and grammar errors have been corrected.
